# High Prevalence of Multidrug-Resistant *Klebsiella pneumoniae* in a Tertiary Care Hospital in Ethiopia

**DOI:** 10.3390/antibiotics10081007

**Published:** 2021-08-20

**Authors:** Tewachew Awoke, Brhanu Teka, Aminu Seman, Shemse Sebre, Biruk Yeshitela, Abraham Aseffa, Adane Mihret, Tamrat Abebe

**Affiliations:** 1Department of Medical Laboratory Sciences, College of Medicine and Health Sciences, Bahir Dar University, Bahir Dar 79, Ethiopia; tewaslike@gmail.com; 2Department of Microbiology, Immunology and Parasitology, School of Medicine, College of Health Sciences, Addis Ababa University, Addis Ababa 9086, Ethiopia; aminu.seman@aau.edu.et (A.S.); shemse.sebre@aau.edu.et (S.S.); adane.mihret@aau.edu.et (A.M.); tamrat.abebe@aau.edu.et (T.A.); 3Bacteriology Unit, Armauer Hansen Research Institute, Addis Ababa 1005, Ethiopia; biruky01@gmail.com (B.Y.); aseffaa@gmail.com (A.A.)

**Keywords:** *K. pneumoniae*, antimicrobial susceptibility patterns, MDR, Ethiopia

## Abstract

*Klebsiella pneumoniae* poses an urgent public health threat, causing nosocomial outbreaks in different continents. It has been observed to develop resistance to antimicrobials more easily than most bacteria. These days, multidrug-resistant strains are being increasingly reported from different countries. However, studies on the surveillance of multidrug-resistant *Klebsiella pneumoniae* are very rare in Ethiopia. This study aimed to determine the antimicrobial resistance patterns and magnitude of MDR *K. pneumoniae* isolates from patients attending or admitted to Tikur Anbessa Specialized Hospital (TASH). A cross-sectional study was conducted from September 2018 to February 2019 at TASH, Addis Ababa, Ethiopia. Identification of *K. pneumoniae* was done by examining the Gram stain, colony characteristics on MacConkey agar and 5% sheep blood agar, as well as using a series of biochemical tests. Antimicrobial susceptibility testing of the isolates for 21 antimicrobials was done by the Kirby–Bauer disc diffusion technique. Data were double entered using Epidata 3.1 and exported to SPSS version 25 software for analysis. Among the total *K. pneumoniae* isolates (*n* = 132), almost all 130 (98.5%) were MDR. Two (1.5%) isolates showed complete non-susceptibility to all antimicrobial agents tested. Moreover, a high rate of resistance was observed to cefotaxime and ceftriaxone 128 (97%), trimethoprim-sulfamethoxazole 124 (93.9%), and cefepime 111 (84.1%). High susceptibility was recorded to amikacin 123 (93.2%), imipenem 107 (81.1%), meropenem 96 (72.7%), and ertapenem 93 (70.5%). *K. pneumoniae* isolates showed a high rate of resistance to most of the tested antimicrobials. The magnitude of MDR *K. pneumoniae* was very alarming. Therefore, strengthening antimicrobial stewardship programs and antimicrobial surveillance practices is strongly recommended in TASH.

## 1. Introduction

*Klebsiella pneumoniae* is a Gram-negative, rod-shaped bacterium that belongs to the family Enterobacteriaceae. *K. pneumoniae* is primarily an opportunistic pathogen that attacks immune-compromised individuals who are hospitalized and suffer from severe underlying diseases. In these patients, it results in hospital-acquired infections associated with the urinary tract, blood-stream, wounds, and respiratory tract [1]. Besides this, *K. pneumoniae* has emerged as a cause of severe community-acquired infections, such as community-acquired pneumonia, pyogenic liver abscess, and metastatic infections such as meningitis [2]. It accounts for about one-third of all Gram-negative infections overall [3].

Antimicrobial resistance is a growing problem in modern healthcare around the world. It is estimated that antimicrobial resistance-related deaths each year will rise from currently 700,000 to 10 million and cost 100 trillion dollars to global economic output by 2050 if containment of antimicrobial resistance at a global level is not effectively implemented [4]. Multidrug-resistant (MDR) bacterial infections that pose a serious risk to patients are increasing worldwide [5]. One of the most common species of bacteria that cause problems in healthcare today is *K. pneumoniae* [6], which, together with other highly important MDR pathogens, comprises the ESKAPE group that stands for *Enterococcus faecium*, *Staphylococcus aureus*, *Klebsiella pneumoniae*, *Acinetobacter baumannii*, *Pseudomonas aeruginosa*, and *Enterobacter* species, to emphasize that these pathogens effectively “escape” the effects of antibacterial drugs [7]. Its contribution to the antimicrobial resistance crisis is impossible to quantify. It is unique amongst the Gram-negative ESKAPE pathogens because of its high diversity of acquired antimicrobial resistance genes, high plasmid load, significantly more varied DNA composition reflecting diverse horizontal gene transfer partners, and broad ecological range [3]. Nowadays, *K. pneumoniae* strains are recognized as an urgent threat to human health, because of the emergence of MDR strains associated with hospital outbreaks [6]. It has gained attention worldwide, especially in developed countries, due to its high drug resistance. Antimicrobial resistance rates in *K. pneumoniae* have steadily increased over the years, and *K. pneumoniae* is becoming resistant to virtually all aminoglycosides, quinolones, and β-lactams [8]. For instance, at the European level, more than one-third (36.6%) of *K. pneumoniae* isolates reported to the European antimicrobial resistance surveillance network for 2019 were resistant to at least one of the antimicrobial groups under regular surveillance, i.e., fluoroquinolones, third-generation cephalosporins, aminoglycosides, and carbapenems [9]. The Ethiopian annual antimicrobial surveillance report showed that 95.8% of *K. pneumoniae* isolates were resistance to ceftriaxone, 86.7% to ceftazidime, 95.6% to trimethoprim-sulfamethoxazole, 83.3% to cefepime, 62.7% to gentamicin, 48.1% to ciprofloxacin, 30.6% to meropenem, and 7.2% to amikacin [10]. *K. pneumoniae* has been observed to develop resistance to antimicrobials more easily than most bacteria [2,11]. One of the main mechanisms of resistance is through β-lactamase production that hydrolyzes β-lactam antibiotics. The most reported β-lactamase genes among *K. pneumoniae* worldwide were *bla*_SHV-12_, *bla*_CTX-M-2_, and *bla*_SHV-5_. Furthermore, occurrence of the *bla*_NDM-1_ gene carrying *K. pneumoniae* has been reported in all continents [12]. In Ethiopia, in a recent systematic review and meta-analysis, the pooled estimates of ESBL-producing *K. pneumoniae* was 64.3% (95% CI: 47.0–81.5) [13]. Few treatment options, such as polymyxins and tigecycline, are available for infections caused by MDR *K. pneumoniae.* However, there are increasing reports of *K. pneumoniae* isolates resistance to these drugs [3]. Thus, the best option is to control the development and spread of antimicrobial resistance. It is obvious that to better understand the extent of drug resistance in different settings and design better control strategies, research on such a type of pathogen is crucial. Continuous global dissemination of multidrug-resistant and extremely drug-resistant *K. pneumoniae,* with superior ability to cause multi-continent outbreaks, have been noted [3,6]. Some of *K. pneumoniae* strains are resistant to about 95% of antimicrobials of the pharmaceutical market in the world [12]. However, to the best of our knowledge, studies emphasizing the degree of drug resistance in *K. pneumoniae* in Africa, particularly in Ethiopia, are still limited. Therefore, this study was planned to determine the antimicrobial-resistance patterns and magnitude of MDR *K. pneumoniae* at Tikur Anbessa Specialized Hospital (TASH), Addis Ababa, Ethiopia. TASH is the largest teaching and referral hospital in Ethiopia.

## 2. Results

### 2.1. Socio-Demographic and Clinical Characteristics 

A total of 132 non-repetitive *K. pneumoniae* isolates collected from patients who were admitted to or attended different departments of TASH were included. As shown in Table 1, among the total isolates, 83 (62.9%) were isolated from males. The majority of the isolates were recovered from blood specimens (63, 47.7%) followed by urine (34, 25.8%), wounds (21, 15.9 %), body fluids (10, 7.6 %), and sputum specimens (4, 3.0%). The average age of the study participants was 13.4 years, varying from at least 1 day to a maximum of 86 years, of which 74 (56.1%) participants were below 5 years. The majority of *K. pneumoniae* isolates were recovered from hospitalized patients (120, 90.9%), of which the largest number were from patients admitted in pediatric wards (53, 44.2%) and intensive care units (ICUs) (46, 38.3%). Conversely, only two isolates were from patients in orthopedics, and two were from emergency wards.

### 2.2. Antimicrobial Susceptibility Patterns of K. pneumoniae Isolates

In this study, the susceptibility of 132 *K. pneumoniae* isolates to 21 antimicrobial agents belonging to 12 antimicrobial categories was determined. As presented in Table 2**,** a high rate of resistance was observed to commonly used antimicrobials such as cefotaxime (128/132, 97%), ceftriaxone (128/132, 97%), trimethoprim-sulfamethoxazole (124/132, 93.9%), and cefepime (111/132, 84.1%). Besides this, a significant intermediate level of resistance was noted to tobramycin (26.5%), amoxicillin-clavulanate (24.2%), ceftazidime (21.2%), piperacillin-tazobactam (20.5%), aztreonam (22.7%), and ciprofloxacin (18.9%). On the other hand, *K. pneumoniae* isolates showed the highest susceptibility to amikacin (123, 93.2%), followed by carbapenem antimicrobials, which were imipenem (107, 81.1%), meropenem (96, 72.7%), and ertapenem (93, 70.5%).

### 2.3. Magnitude of Multidrug Resistance among K. pneumoniae Isolates

Almost all (130/132, 98.5%) of the isolates were non-susceptible to at least three antimicrobials in different categories and, hence, defined as MDR, whereas only two (1.5%) isolates were not MDR. In our case, the MDR range was wide, which encompassed resistance to three (R3) to 12 (R12) antimicrobial categories. Consequently, it was necessary to indicate the number of isolates in each level of MDR. Based on this, from the total isolates, 20/132 (15.2%) showed non-susceptibility to at least one antimicrobial agent in all categories. Of these, two isolates showed complete non-susceptibility to all antimicrobial agents. Furthermore, 16/132 (12.1%) isolates were non-susceptible to at least one antimicrobial agent in 11 antimicrobial categories. Only one isolate was non-susceptible to at least one antimicrobial agent in merely three antimicrobial categories, which was the least MDR (Figure 1). The details of multidrug-resistance patterns of *K. pneumoniae* isolates are presented in Appendix A.

### 2.4. Distribution of MDR K. pneumoniae Isolates among Inpatient and Outpatient Wards

From the total MDR *K. pneumoniae* isolates, the large majority were isolated from patients in pediatric wards (52/130, 40%) and ICUs (46/130, 35.4%), followed by the outpatient department (11/130, 8.5%). The lowest proportion (4/130, 3.1%) of MDR isolates was obtained from those in orthopedics and emergency wards (Figure 2). 

## 3. Discussion

*Klebsiella pneumoniae* is one of the multidrug resistant microorganisms identified as an urgent threat to human health by the World Health Organization [5] and Centers for Disease Control and Prevention [14]. It is rapidly becoming untreatable using even last-line antimicrobials, especially in hospital settings [6], suggesting the need for continuous surveillance of antimicrobial resistance patterns. This work focused on the magnitude of antimicrobial resistance among clinical *K. pneumoniae* isolates.

### 3.1. Antimicrobial Resistance Patterns of K. pneumoniae Isolates

Third-generation cephalosporins have been the treatment of choice for Gram-negative bacteria, including *K. pneumoniae,* but nowadays, they are largely ineffective due to the emergence of ESBL-producing bacteria [8]. This was also true in our study, in which one of the high antimicrobial resistance was against 3rd and 4th generation cephalosporins, ceftriaxone (97%), cefotaxime (97%), cefepime (84.1%), and ceftazidime (65.9%). Likewise, high resistance rates to these antimicrobials were also observed in other recent studies in Ethiopia: 86.4% to cefotaxime and 85.4% to ceftazidime and cefepime in Addis Ababa [15] and 97.6% to ceftazidime, 94.1% to cefepime, and 88.2% to ceftriaxone in Bahir Dar [16]. However, Badamchi et al. from Iran reported a resistance rate of 60.2% to ceftriaxone, 32.6% cefotaxime, and 54.8% cefepime, which was lower than the results of our study [17].

In this study, more than half of the participants had taken 3rd and 4th generation cephalosporins prior to recruitment to the study (see Appendix A). Additionally, a previous study by Gutema et al. found that three out of four patients in the hospital where we conducted our study were prescribed antimicrobials empirically, of whom almost 90% were prescribed broad-spectrum β-lactams [18]. Therefore, the high resistance to broad-spectrum β-lactams might be attributed to the indiscriminate use of these antimicrobials, which creates a selective pressure. Resistance to trimethoprim-sulfamethoxazole was 93.9%, which is comparable with different studies in Ethiopia; 86.4% in Addis Ababa [15] and 96.5% in Bahir Dar [16]; furthermore, there was a consistent report from Tunisia citing 94.06% [19]. Nevertheless, it was higher than those observed in studies conducted in Bangladesh (44%) [20] and Nepal (51.3%) [21]. High resistance to trimethoprim-sulfamethoxazole may be due to variations in prescription practices and self-medication because of affordability and availability [22]. The main mechanism of resistance to trimethoprim-sulfamethoxazole in *K. pneumoniae* is the acquisition of transferable *dfr* genes, which could mediate the overproduction of dihydrofolate reductase and the presence of drug-resistant dihydropteroate synthetase enzymes, encoded by *sul* genes [23].

In the current study, from all tested antimicrobials, the best susceptibility (93.2%) was observed to amikacin. A low resistance of *K. pneumoniae* to amikacin was also reported in previous study in Ethiopia [15]. The lower resistance to amikacin may be due to its absence of use as empirical therapy and non-existence of considerable cross-resistance with β-lactam antimicrobials [24]. 

*K. pneumoniae* isolates were more susceptible to carbapenems, following to amikacin, with an overall non-susceptibility rate of 29.6%. Imipenem was the most active carbapenem and revealed a susceptibility rate of 81.1%, followed by meropenem (72.7%) and ertapenem (70.5%). This is in accordance with a previous study conducted in Egypt, which reported susceptibility rates of 75% to meropenem and imipenem [25]. In a study in Addis Ababa, Ethiopia [15], 10.7% of *K. pneumoniae* isolates were resistant to meropenem, which was lower than our study. Moreover, there was also a study in which no resistance was detected to carbapenems [19], but our study showed a considerable carbapenem resistance, considering that carbapenems are the last-resort antibiotics. In the current study, 15.2% of the study participants had taken carbapenems prior to recruitment to the study (see Appendix A). Empirical prescription of carbapenems, particularly meropenem, was very common in the hospital [26], which may have created resistance and furthered the emergence of carbapenemase-producing bacteria. 

### 3.2. Multidrug Resistance among K. pneumoniae Isolates

In our study, almost all (98.5%) *K. pneumoniae* isolates were MDR, which is comparable with the result of a study conducted in Gondar, Ethiopia (citing 95.6%) from patients with urinary tract infections [27] and Iraq (citing 100%) [28]. On the contrary, our result was higher than studies conducted in Addis Ababa, (citing 83.5%) [15] and Bahir Dar (citing 87.6%) [16]. This high prevalence might be because the majority (90.91%) of *K. pneumoniae* isolates were collected from hospitalized patients, where many antimicrobials were being prescribed in the hospital, which serves as a selective pressure for drug resistance. Moreover, in this study, more antimicrobial agents were covered in the assessment based on the recommendations of Magiorakos et al., which increases the detection of MDR [29]. 

Generally, in our study, *K. pneumoniae* showed a high resistance to most of the antimicrobials tested, which is very alarming. A study on *K. pneumoniae* in five African and two Vietnamese major towns stated uncontrolled consumption of antimicrobial agents through self-medication, inappropriate antibiotic prescription, the substandard quality of some drugs, and a lack of effective measures to prevent nosocomial infections as key factors for facilitating the spread of antimicrobial resistance among studied countries [30]. Since these conditions could be present in Ethiopia, the high drug resistance observed in the current study could also be due to these factors.

## 4. Materials and Methods

### 4.1. Study Design

A cross-sectional study was conducted on a total of 132 study participants. *K. pneumoniae* were isolated from each of the study participants from September 2018 to February 2019. Participants were enrolled using a convenient sampling technique. Socio-demographic characteristics were obtained using a well-designed questionnaire.

### 4.2. Bacterial Isolation and Identification

Biological specimens were inoculated on appropriate culture media. Briefly, blood specimens were collected on BacT/ALERT^®^ 3D culture bottles and incubated in automatic BacT/ALERT^®^ 3D at 35–37 °C in 5% CO_2_ for 5 days or until they signaled positive for growth. The microbial growth that could be detected by the flag and audible sound of the instrument was subsequently sub-cultured on MacConkey agar (Oxoid, Basingstoke, UK), 5% sheep blood agar (Oxoid, UK), and chocolate agar (Oxoid, UK) plates. A negative result was reported by examining the flag, gram staining, and subculturing at the end of the 5^th^ day before discarding it as negative [31]. Specimens from wound, sputum, and body fluids were cultured on MacConkey agar, 5% sheep blood agar, and chocolate agar, while inoculation of urine was done only on MacConkey agar and 5% sheep blood agar. After incubation of all inoculated culture plates at 35–37 °C for 18–24 hours, preliminary identification of *K. pneumoniae* was done through examining colony characteristics on MacConkey agar and blood agar. Further identification was done using a Gram stain and a series of biochemical tests, including indole, triple sugar iron agar, citrate, mannitol, malonate, lysine decarboxylase, urea, and motility medium. *K. pneumoniae* is Gram-negative and rod shaped, indole negative, a gas and acid producer, hydrogen sulfide negative, citrate positive, a mannitol fermenter, malonate positive, lysine decarboxylase positive, urea slow producing, and non-motile [2,32]. After identification, the isolates were sub-cultured on 5% sheep blood agar plate and incubated overnight to get fresh colonies for antimicrobial susceptibility testing (AST).

### 4.3. Antimicrobial Susceptibility Testing

Using a sterile wire loop, 3–5 pure colonies were picked from blood agar and emulsified in 3–4 mL normal saline to prepare a 0.5 McFarland standard using a McFarland Densitometer. From the preparation, we inoculated onto Muller–Hinton agar (Oxoid, UK) using a sterile swab for the AST [33]. The AST was performed using the Kirby–Bauer disc diffusion method, with the following antimicrobial discs: tetracycline (30 μg), gentamicin (10 μg), amikacin (30 μg), tobramycin (10 μg), ciprofloxacin (5 μg), nalidixic acid (30 μg), amoxicillin-clavulanate (20/10 μg), piperacillin-tazobactam(100/10 μg), trimethoprim-sulfamethoxazole (1.25/23.75 μg), aztreonam (30 μg), chloramphenicol (30 μg), cefoxitin (30 µg), Cefazolin (30 μg), Cefuroxime (30 μg), Ceftriaxone (30 μg), Ceftazidime (30 μg), cefotaxime (30 μg), cefepime (30 µg), imipenem (10 µg), ertapenem (10 μg), and meropenem (10 µg) (Oxoid, Basingstoke, UK and BD, Franklin Lakes, NJ, USA). The results were read and interpreted after 16–18 h of incubation at 35 ± 2 °C [34]. Control strains *K. pneumoniae* ATCC^®^ 700603 were used during identification, as well as *Escherichia coli* ATCC^®^ 25922, *Pseudomonas aeruginosa* ATCC^®^ 27853 (for carbapenems), *Escherichia coli* ATCC^®^ 35218 (for β-lactam/ β-lactamase inhibitor combinations) for controlling the potency of the drugs. MDR was defined as non-susceptibility to at least one agent in three or more antimicrobial categories [29].

### 4.4. Data Analysis

Data were checked, cleaned, and double entered into Epidata software version 3.1 (The EpiData Association, Enghavevej, Odense, Denmark), and then it was exported to Statistical Package for Social Sciences (SPSS version 25.0, IBM Corp., Armonk, NY, USA) software for analysis. Descriptive statistics were used to present the findings.

### 4.5. Ethics Approval and Consent to Participate

This study was approved by the Ethics Review Committee of the Department of Microbiology, Immunology and Parasitology, School of Medicine, College of Health Sciences, Addis Ababa University (Reference number: DERC/17/18/02-N) and the AHRI/ALERT ethical review committee (Protocol number: PO12/18). A permission letter was obtained from TASH. Moreover, before commencing the study, a written informed consent/assent was obtained from each study participant. Confidentiality was maintained for all data collected.

## 5. Conclusions

The findings of our study revealed that *K. pneumoniae* isolates showed a high resistance to most of the drugs commonly used to treat infections, and the magnitude of MDR *K. pneumoniae* isolates was very alarming. Virtually all (98.5%) of the isolates were MDR. Higher susceptibility was observed to amikacin and carbapenems. Therefore, our data should be used as an alert for the need for prevention and control of MDR *K. pneumoniae* in hospital settings, most specifically within the study area. 

## Figures and Tables

**Figure 1 antibiotics-10-01007-f001:**
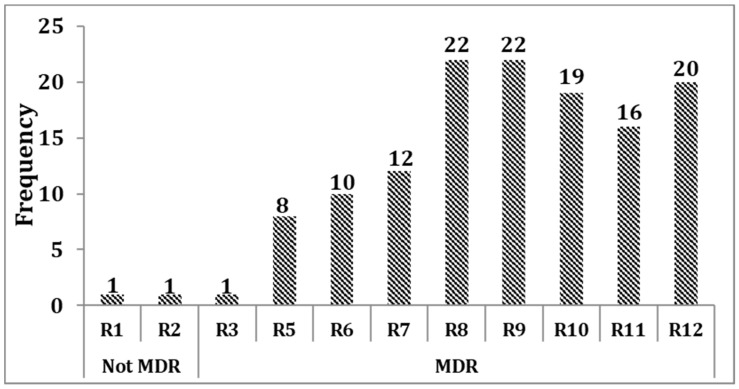
Multidrug resistance levels of *K. pneumoniae* isolates. MDR: multidrug-resistance; Rn: non-susceptibility to at least one antimicrobial agent in “*n*” antimicrobial categories, where “*n*” is the number of antimicrobial categories.

**Figure 2 antibiotics-10-01007-f002:**
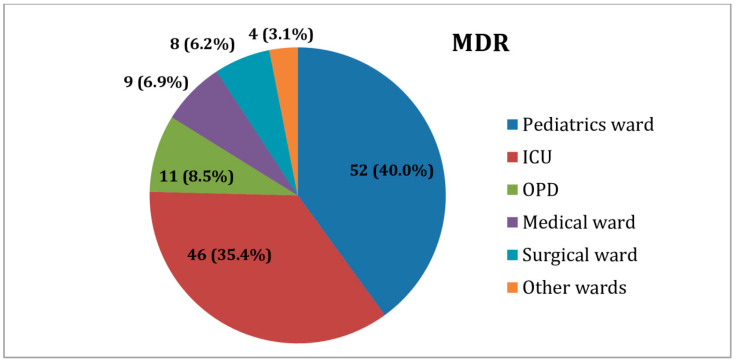
Distribution of MDR *K. pneumoniae* isolates among inpatients and outpatients. ICUs: Intensive care units; other wards: emergency (2) and orthopedics (2) OPD: outpatient department.

**Table 1 antibiotics-10-01007-t001:** Distribution of *K. pneumoniae* isolates among demographic and clinical characteristics of study participants.

**Variables**	**Category**	**Frequency**	**Percentage**
**Gender**	Male	83	62.9
	Female	49	37.1
**Age**	birth to <5 years	74	56.1
	5 years to <18 years	20	15.2
	18 years to <45 years	25	18.9
	≥45 years	13	9.8
**Specimen Type**	Blood	63	47.7
	Urine	34	25.8
	Wound	21	15.9
	Body fluids	10	7.6
	Sputum	4	3.0
**Patient Setting**	Inpatient	120	90.9
	Outpatient	12	9.1
**Ward type**	Pediatric ward	53	44.2
	ICU	46	38.3
	Medical ward	9	7.5
	Surgical ward	8	6.7
	Other wards	4	3.3

Other wards: emergency (2) and orthopedics (2).

**Table 2 antibiotics-10-01007-t002:** Antimicrobial susceptibility patterns of *K. pneumoniae* isolates.

**Antimicrobial Categories**		**Antimicrobial Agents**	**Antimicrobial Susceptibility Pattern**		
			**Susceptible** ***n*** **(%)**	**Inter** **Mediate** ***n*** **(%)**	**Resistant** ***n*** **(%)**
Tetracyclines		Tetracycline	21(15.9)	15(11.4)	96(72.7)
Aminoglycosides		Gentamicin	29(22.0)	8(6)	95(72.0)
		Tobramycin	49(37.1)	35(26.5)	48(36.4)
		Amikacin	123(93.2)	5(3.8)	4(3.0)
Fluoroquinolones		Ciprofloxacin	58(43.9)	25(18.9)	49(37.1)
		Naldixic acid	30(22.7)	42(31.8)	60(45.5)
Antipseudomonal Penicillins + β-lactamase inhibitors		Piperacillin-tazobactam	55(41.7)	27(20.5)	50(37.9)
Penicillins + β-lactamase inhibitors		Amoxicillin-clavulanate	18(13.6)	32(24.2)	82(62.1)
Folate pathway inhibitors		Trimethoprim-Sulfamethoxazole	6(4.5)	2(1.5)	124(94)
Phenicols		Chloramphenicol	59(44.7)	13(9.8)	60(45.5)
Cephamycins		Cefoxitin	62(47.0)	12(9.1)	58(43.9)
β-lactams	1st and 2nd generation cephalosporins	Cefuroxime	2(1.5)	2(1.5)	128(97.0)
		Cefazolin	1(0.8)	1(0.8)	130(98.5)
	3rd and 4th generation cephalosporins	Cefepime	5(3.8)	16(12.1)	111(84.1)
		Ceftriaxone	4(3.0)	0(0.0)	128(97.0)
		Cefotaxime	4(3.0)	0(0.0)	128(97.0)
		Ceftazidime	17(12.9)	28(21.2)	87(65.9)
	Monobactams	Aztreonam	13(9.8)	30(22.7)	89(67.4)
	Carbapenems	Meropenem	96(72.7)	4(3.0)	32(24.3)
		Imipenem	107(81.1)	9(6.8)	16(12.1)
		Ertapenem	93(70.5)	5(3.7)	34(25.8)

## Data Availability

The datasets supporting the conclusions of this article are included within the article and its additional files.

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
