# Peer review of "High Prevalence of Multidrug-Resistant Klebsiella pneumoniae in a Tertiary Care Hospital in Ethiopia"

_antibiotics, 2021, doi:10.3390/antibiotics10081007_

Round 1
Reviewer 1 Report
Awoke et al. aimed to investigate the antimicrobial resistance of 132 K. pneumoniae isolates from Tikur Anbessa Specialized Hospital, Ethiopia. Species determination was performed by biochemical tests and Gram staining. Antimicrobial susceptibility to 21 compounds was studied by the Kirby-Bauer method. 130 K. pneumoniae isolates were categorized as MDR. High rates of resistance were found to several beta-lactams and trimethoprim-sulfamethoxazole.
Major comments:
This reviewer does not find much novelty in the data presented, except that it may be of interest to local authorities. A study covering more hospitals in the country would be more relevant.
The interest of the work would be considerably enhanced with a study of the clonality of the strains (MLST). It would also be relevant to know the mechanism of resistance to the main groups of antibiotics. For example, the identification of beta-lactamases (main mechanisms of resistance to beta-lactams) or qnR genes (involved in resistance to quinolones), among others.
I also recommend a review of the article by a mother tongue, some of the expressions used are not entirely correct.
Minor comments:
Introduction:
- Line 37. Change “the bacterium” by “K. pneumoniae”. The term "the bacterium" is overused in the introduction. I suggest changing it to "K. pneumoniae" or "K. pneumoniae strains".
- Line 42. Add “and” between “abscess” and “metastasic”.
- Line 62-64. Recent examples of increasing resistance to aminoglycosides, quinolones, and beta-lactams should be given. For example, the distribution of KPC or VIM enzymes related to increased beta-lactam resistance.
- Line 64-65. Provide a reference for this sentence.
- Line 65-70. Revise and rewrite these lines.
Materials and Methods:
- Line 100. The determination of the bacterial species by biochemical tests is currently in disuse due to the use of MALDI-TOF or VITEK-2. Please mention the biochemical tests performed for the identification.
- Colistin, an antibiotic of last resort, is increasingly used against K. pneumoniae MDR strains. It would be interesting to know the antimicrobial activity of this antibiotic against the 132 strains.
Results
- Line 139. The number of patients should be indicated.
Discussion
- Section 4.1 shows different font sizes and styles. Please correct it.
- Line 260. Indicate in one sentence the main mechanism of resistance to trimethoprim-sulfamethoxazole in K. pneumoniae.
Tables
- Table 2 (page 5). Group all beta-lactam antibiotics into one category. Monobactams, carbapenems, etc. indicate as a subcategory within beta-lactam antibiotics.
Figures
- Figure 2 is difficult to understand. I suggest adding as supplementary material an Excel file with the susceptibility results for each of the 132 strains. Clinical and demographic information could also be included in this file.
- Figure 3. Correct the footnote. The intensive care unit is out of place..
Reviewer 2 Report
The article is quite interesting; however, before it could be considered for its possible publication, I have some major quires which authors need to incorporate and extensively revise their manuscript.
General comments Sentence formation needs crosscheck. Grammatical mistakes need to be minimized. English needs to be checked and corrected by some native English speaker.
Abstract section does not give proper information. Why study was performed on Klebsiella pneumoniae and why authors have not included other microorganisms like K. Georgiana. How authors have narrowed down to Klebsiella pneumoniae based nosocomial outbreaks in different continents. Better to mention few countries and references of works performed on Klebsiella pneumoniae.
Authors have checked resistance towards different antibiotics and found isolates MDR. Have they checked resistance for Colistin? Do authors found some isolates that is progressing to XDR phenotype.
Authors have mentioned that “Little is known concerning the magnitude of multidrug resistant (MDR) K. pneumoniae in Ethiopia”, what about infections caused by other microorganisms. Please revise the abstract (needs to rephrase and rewrite some sentences). Also, highlight essentialities and future perspectives of the study.
Section Introduction
Authors need to give a background of the rising antibiotic resistance problem from Ethiopia. Most reported the studies going on in different parts of the world, but have reported none that could have been reported for K. pneumoniae or any other microbial isolates. Additionally, authors have target isolates that tend to be categorized as MDR, but the problem is more serious with XDR phenotype in the hospital settings.
Section Material methods
Study design: Participants who were culture positive for K. pneumoniae. What do you mean by culture-positive individuals? Please rephrase the sentence so as to bring clarity to the thoughts.
Bacterial isolation and identification: What biochemical tests and procedure was adopted for testing. Why identification based on 16SrRNA was not performed. It is necessary to identify them using sequencing and submit the same in some public domain databases such as NCBI.
Section Results
Why biofilm formation was not studied. Have the authors noticed the type of resistant determinants such as CTX-M and others? It is better to perform a study on the type of variants that impart resistance to these isolates in hospital settings. For idea, check information on Hemlata et al., 2021, Microbial Pathogenesis; Azam et al., 2016 Front Micrbiol.
Conclusion
The sections are needed to be re-written and updated for summarizing information present in the paper. I find only a few references from 2019, 2020, and 2021.
Round 2
Reviewer 1 Report
I consider that the work has improved slightly, the authors have satisfactorily addressed all the minor comments I indicated in my previous review. However, the clonality of the strains, the most important resistance mechanisms of each strain, or the susceptibility to relevant antibiotics (especially colistin) remain unexamined. I believe this is a huge limitation of the work.
Reviewer 2 Report
The authors have addressed most of my comments.
